# Determinants of Longitudinal Change of Glycated Hemoglobin in a Large Non-Diabetic Population

**DOI:** 10.3390/jpm11070648

**Published:** 2021-07-08

**Authors:** Ho-Ming Su, Wen-Hsien Lee, Ying-Chih Chen, Yi-Hsueh Liu, Jiun-Chi Huang, Pei-Yu Wu, Szu-Chia Chen

**Affiliations:** 1Department of Internal Medicine, Kaohsiung Municipal Siaogang Hospital, Kaohsiung 812, Taiwan; cobeshm@seed.net.tw (H.-M.S.); cooky-kmu@yahoo.com.tw (W.-H.L.); 990329kmuh@gmail.com (Y.-C.C.); liuboy17@gmail.com (Y.-H.L.); karajan77@gmail.com (J.-C.H.); wpuw17@gmail.com (P.-Y.W.); 2Division of Cardiology, Department of Internal Medicine, Kaohsiung Medical University Hospital, Kaohsiung 807, Taiwan; 3Faculty of Medicine, College of Medicine, Kaohsiung Medical University, Kaohsiung 807, Taiwan; 4Division of Nephrology, Department of Internal Medicine, Kaohsiung Medical University Hospital, Kaohsiung Medical University, Kaohsiung 807, Taiwan; 5Research Center for Environmental Medicine, Kaohsiung Medical University, Kaohsiung 807, Taiwan

**Keywords:** non-diabetes, glycated hemoglobin, longitudinal change

## Abstract

Although many cross-section studies have assessed the determinants of glycated hemoglobin (HbA_1c_), there have been limited studies designed to evaluate the temporal correlates of HbA_1c_ in non-diabetic patients. This study aimed to identify the major determinants of longitudinal change of HbA_1c_ in non-diabetic patients. This study included subjects from the 104,451 participants enrolled between 2012 and 2018 in the Taiwan Biobank. We only included participants with complete data at baseline and follow-up (*n* = 27,209). Patients with diabetes at baseline or follow-up (*n* = 3983) were excluded. Finally, 23,226 participants without diabetes at baseline and follow-up were selected in this study. △Parameters was defined as the difference between the measurement baseline and follow-up. Multivariable linear regression analysis was used to identify the major determinants of HbA_1c_ longitudinal change (△HbA_1c_). During a mean 3.8 year follow-up, after multivariable analysis, new-onset hypertension (coefficient β: 0.014, *p* < 0.001), high △heart rate (coefficient β: 0.020, *p* = 0.002), high △BMI (coefficient β: 0.171, *p* = 0.028), high △fasting glucose (coefficient β: 0.107, *p* < 0.001), low △creatinine (coefficient β: −0.042, *p* < 0.001), high △total cholesterol (coefficient β: 0.040, *p* < 0.001), high △hemoglobin (coefficient β: 0.062, *p* < 0.001), high △GPT (coefficient β: 0.041, *p* = 0.001), and low △albumin (coefficient β: −0.070, *p* < 0.001) were significantly associated with high △HbA_1c_. In non-diabetic population, strategies to decrease the development of new-onset hypertension, resting heart rate, body mass index, fasting glucose, total cholesterol, and GPT and increase serum albumin level might be helpful in slowing the longitudinal change of HbA_1c_. In addition, increased hemoglobin and decreased serum creatinine over time also had an impact on the HbA_1c_ elevation over time in non-diabetic population.

## 1. Introduction

Glycated hemoglobin (HbA_1c_) is a useful biomarker indicating mean blood glucose concentrations over the past two to three months and has an important role in the evaluation of glucose control and predicts future adverse cardiovascular events [1,2,3,4]. HbA_1c_ measurement may be included in the health examination program for the identification of diabetes [5].

HbA_1c_ is not only a helpful diagnostic and prognostic parameter in patients with diabetes, but is also useful in predicting future diabetic risk in non-diabetic patients [6]. Ewid et al. reported that HbA_1c_ could be used as a predictive biomarker for coronary artery disease in non-diabetic low-risk patients [7]. Arbel et al. found in non-diabetic patients with myocardial infarction or stable angina, the HbA_1c_ level was correlated with the severity of coronary artery disease as measured by the SYNTAX score. However, no correlation was found between admission glucose or fasting glucose levels and severity of coronary artery disease [8]. In addition, an upper-normal HbA_1c_ level was associated with cardiovascular and overall mortality in non-diabetic hemodialysis patients, whereas a lower HbA_1c_ level was not [9].

Many cross-section studies ahve assessed the relationship between clinical important parameters and glucose metabolism in non-diabetic patients. Higher resting heart rate in individuals without diabetes was reported to be associated with future unfavorable changes in insulin levels and insulin sensitivity. Resting heart rate might be a risk marker for future pathophysiological changes in glucose metabolism [10]. Systolic and diastolic blood pressures were useful markers in predicting future diabetes in non-diabetic patients [11]. HbA_1c_ concentration was found to be accompanied with an increase in body mass index (BMI) regardless of diabetes status [12]. A U-shaped relationship between fasting plasma glucose and serum uric acid levels existed in Chinese individuals with normal glucose tolerance [13]. High cholesterol and triglyceride have been shown to be closely correlated to high fasting plasma glucose in non-diabetic patients [14]. Previous studies have demonstrated a relationship between elevated liver enzymes (particularly transaminases) and incident type 2 diabetes [15]. Kang et al. included 24,594 non-diabetic subjects and found that HbA_1c_ increased as renal function declined [16]. However, there were only limited studies to evaluate the relationship between longitudinal changes of above clinical important parameters and glucose metabolism in non-diabetic patients. The temporal association of above clinical important parameters and glucose metabolism over time has not been studied well. Discovering the longitudinal association between changes in above clinical important parameters and changes in glucose metabolism over time in large-scale studies should be more informative than in cross-sectional studies. Hence, this study aimed to examine the longitudinal association of clinical important parameters (including age, presence of hypertension, blood pressure, resting heart rate, BMI, uric acid, lipid profile, and renal and liver function) with HbA_1c_ in a large cohort study with repeated measurements of above data over a mean of 3.8 years of follow up and identify the major determinants of longitudinal change of HbA_1c_ in non-diabetic patients.

## 2. Methods

### 2.1. Data Source and Study Population

The data contained in this study were collected from the Taiwan Biobank (TWB), a general population-based research database comprised of cancer-free residents aged 30−70 years included from 31 recruitment stations in Taiwan since 2008. The data source of TWB has been described previously [17,18]. The methodologies of data collection from all participants in TWB were the same and had a standardized procedure. Details regarding the laboratory data collection and questionnaire of TWB English version can be found from its official website (https://www.twbiobank.org.tw/new_web_en/; accessed on 14 April 2020). Written informed consent was acquired from all enrolled participants and all investigations in this study were conducted according to the Declaration of Helsinki. The present study was approved by the Institutional Review Board of Kaohsiung Medical University Hospital (KMUHIRB-E(I)−20180242), which was approved on 8 March 2018.

The present study included study subjects from the 104,451 participants enrolled between 2012 and 2018 in TWB. The demographic information including age, gender, and history of diabetes and hypertension, was acquired from a face-to-face interview with TWB investigators. BMI, systolic and diastolic blood pressures, resting heart rate, and overnight fasting blood chemistry parameters including fasting glucose, total cholesterol, triglycerides, uric acid, serum creatinine, hemoglobin, glutamic-oxalocetic transaminase (GOT), glutamic-pyruvic transaminase (GPT), albumin, and HbA_1c_ is not only a helpful diagnostic and prognostic parameter in patients with diabetes, but is also useful in predicting future diabetic risk in non-diabetic patients were collected. All of theabove data were obtained at baseline and at around 3.8-year follow-up. We only included participants with complete data at baseline and 3.8-year follow-up (*n* = 27,209). Patients with diabetes at baseline or follow-up (*n* = 3983) were excluded. Finally, 23,226 participants without diabetes at baseline and follow-up (7896 men and 15,430 women) were selected in this study.

### 2.2. Definition of People without Diabetes

Participates who did not take anti-diabetic medication, had no past history of diabetes, and whose fasting glucose was less than 126 mg/dL and HbA_1c_ was less than 6.5% were considered not to have diabetes.

### 2.3. Statistical Analysis

We used SPSS 22.0 for Windows (SPSS Inc., Chicago, IL, USA) to perform statistical analysis. Data were expressed as mean ± standard deviation and percentage. The differences between longitudinal changes of clinical and laboratory data during the 3.8-year follow-up were checked by paired *t*-test. △Parameters was defined as the difference between the measurement baseline and follow-up. Statistically significant variables in univariable linear regression analysis were selected into the multivariable analysis. Multivariable linear regression analysis was used to identify the major determinants of baseline HbA_1c_ and HbA_1c_ longitudinal change (△HbA_1c_). A *p* value of less than 0.05 was considered to be significant.

## 3. Results

Table 1 shows the longitudinal changes of clinical and laboratory data during 3.8-year follow-up in 23,326 study subjects. Compared to baseline data, age, prevalence of hypertension, systolic and diastolic blood pressures, heart rate, BMI, fasting glucose, HbA_1c_, serum creatinine, total cholesterol, triglyceride, GOT, and GPT were increased and uric acid and albumin were decreased at follow-up. 

Table 2 shows the univariable and multivariable correlates of baseline HbA_1c_. After the multivariable analysis, old age (coefficient β: 0.207, *p* < 0.001), female (coefficient β: −0.043, *p* < 0.001), high BMI (coefficient β: 0.096, *p* < 0.001), high fasting glucose (coefficient β: 0.288, *p* < 0.001), low creatinine (coefficient β: −0.030, *p* < 0.001), high uric acid (coefficient β: 0.027, *p* < 0.001), high total cholesterol (coefficient β: 0.128, *p* < 0.001), high triglyceride (coefficient β: 0.031, *p* < 0.001), low GOT (coefficient β: −0.036, *p* < 0.001), high GPT (coefficient β: 0.068, *p* < 0.001), and low albumin (coefficient β: −0.044, *p* < 0.001) were significantly associated with high baseline HbA_1c_. 

Table 3 shows the correlation between HbA_1c_ longitudinal change and longitudinal changes of other parameters in the univariable and multivariable analyses. In the multivariable analysis, new-onset hypertension (coefficient β: 0.014, *p* < 0.001), high △heart rate (coefficient β: 0.020, *p* = 0.002), high △BMI (coefficient β: 0.171, *p* = 0.028), high △fasting glucose (coefficient β: 0.107, *p* < 0.001), low △creatinine (coefficient β: −0.042, *p* < 0.001), high △total cholesterol (coefficient β: 0.040, *p* < 0.001), high △hemoglobin (coefficient β: 0.062, *p* < 0.001), high △GPT (coefficient β: 0.041, *p* = 0.001), and low △albumin (coefficient β: −0.070, *p* < 0.001) were significantly associated with high △HbA_1c_. 

We have further performed two subgroup analyses (group of fasting glucose <100 mg/dL and group of 100 ≤ fasting glucose < 126 mg/dL) to survey the correlation between HbA_1c_ longitudinal change and longitudinal changes of other parameters. In participants with fasting glucose <100 mg/dL (*n* = 20,248), new-onset hypertension (coefficient β: 0.018, *p* = 0.039), low △systolic blood pressure (coefficient β: −0.001, *p* < 0.001), high △diastolic blood pressure (coefficient β: 0.001, *p* < 0.001), high △BMI (coefficient β: 0.034, *p* < 0.001), high △fasting glucose (coefficient β: 0.005, *p* < 0.001), low △creatinine (coefficient β: −0.077, *p* < 0.001), high △total cholesterol (coefficient β: 0, *p* < 0.001), high △hemoglobin (coefficient β: 0.014, *p* < 0.001), high △GPT (coefficient β: 0, *p* = 0.012), and low △albumin (coefficient β: −0.082, *p* < 0.001) were significantly associated with high △HbA_1c_.

In participants with 100 ≤ fasting glucose < 126 mg/dL (*n* = 3078), high △heart rate (coefficient β: 0.001, *p* = 0.009), high △BMI (coefficient β: 0.053, *p* < 0.001), high △fasting glucose (coefficient β: 0.004, *p* < 0.001), low △creatinine (coefficient β: −0.049, *p* = 0.036), low △uric acid (coefficient β: −0.009, *p* = 0.040), high △total cholesterol (coefficient β: 0, *p* = 0.015), high △hemoglobin (coefficient β: 0.029, *p* < 0.001), low △GOT (coefficient β: −0.001, *p* = 0.045), high △GPT (coefficient β: 0.001, *p* = 0.049), and low △albumin (coefficient β: −0.124, *p* < 0.001) were significantly associated with high △HbA_1c_.

## 4. Discussion

In the present study, we evaluated the correlates of HbA_1c_ longitudinal change in 23,326 non-diabetic subjects during a mean 3.8-year follow-up. We found new-onset hypertension and longitudinal changes of heart rate, BMI, fasting glucose, total cholesterol, hemoglobin, and GPT were positively correlated, and longitudinal changes of serum creatinine and albumin were negatively correlated with HbA_1c_ longitudinal change.

Although multivariable analysis showed presence of hypertension and systolic and diastolic blood pressures at baseline had no correlation with baseline HbA_1c_, new-onset hypertension was positively associated with HbA_1c_ longitudinal change after multivariable adjustment. Tsai et al. demonstrated that hypertension was a risk factor of new-onset diabetes [19]. In this study involving non-diabetic patients, we further found patients with new-onset hypertension had a higher HbA_1c_ longitudinal change.

Although baseline resting heart rate had no correlation with baseline HbA_1c_, longitudinal change of heart rate had a positive correlation with HbA_1c_ longitudinal change after multivariable analysis. However, although significant, the HbA_1c_ longitudinal change was small (0.020%) when heart rate increased by 1 beat/min. Kim et al. found an increase in resting heart rate over a two-year follow-up period was significantly associated with a risk of diabetes, independently of baseline resting heart rate and glycol-metabolic parameters in 7,416 adults without diabetes [20]. In this study, we similarly demonstrated resting heart rate change was positively associated with HbA_1c_ change over a 3.8-year follow-up in non-diabetic population. Hence, control of resting heart rate might be beneficial to slow HbA_1c_ longitudinal change.

In the present study, baseline BMI had a strong positive correlation with baseline HbA_1c_ and longitudinal change of BMI had also a strong positive correlation with HbA_1c_ longitudinal change in the multivariable analysis. Maw et al. found higher BMI might have influenced HbA_1c_ trends during one to two-year follow-up [21]. In this study, our result further showed BMI change over time had a strong positive correlation with HbA_1c_ longitudinal change. It suggested good body weight control was effective in slowing HbA_1c_ change in non-diabetic patients.

Baseline fasting glucose had a strong positive correlation with baseline HbA_1c_ and longitudinal change of fasting glucose also had a strong positive correlation with HbA_1c_ longitudinal change in our multivariable analysis, which was consistent with previous studies [22,23,24]. Hence, slowing longitudinal change of fasting glucose should be a useful strategy to reduce HbA_1c_ longitudinal change.

Baseline serum creatinine was positively associated with baseline HbA_1c_ in the univariable analysis, but such association became negative after multivariable adjustment. Hence, baseline serum creatinine was not a major determinant of increased HbA_1c_ at baseline. In addition, longitudinal change of blood creatinine was negatively correlated with HbA_1c_ longitudinal change in our multivariable analysis. Hu et al. examined a prospective association between serum creatinine level and diabetes in 31,343 male workers without diabetes and found low serum creatinine was associated with an increased risk of diabetes [25]. Our present study further demonstrated that serum creatinine decrease over time was associated with HbA_1c_ increase over time during 3.8-year follow-up in non-diabetic population. Hence, decreased blood creatinine over time might reflect decreased skeletal muscle mass over time [25]. Skeletal muscle was a primary target for insulin action [26] and decreased skeletal muscle mass might potentially trigger insulin resistance [27,28], which might cause HbA_1c_ increase over time. 

In this study, baseline uric acid was positively associated with baseline HbA_1c_, which was consistent with the previous study [29]. However, our result demonstrated longitudinal change of uric acid had no significant correlation with HbA_1c_ longitudinal change after multivariable analysis. Hence, longitudinal change of uric acid may only have no or a minor role in HbA_1c_ longitudinal change in our present study. 

Our result showed baseline total cholesterol and triglyceride were positively associated with baseline HbA_1c_, which was similar with previous studies [30,31]. In addition, longitudinal change of total cholesterol had also a positive correlation with HbA_1c_ longitudinal change in our multivariable analysis. Hence, control of total cholesterol should be useful in slowing HbA_1c_ longitudinal change.

Several previous studies reported HbA_1c_ increased with decreasing hemoglobin levels [32,33,34]. However, other studies reported a positive correlation [35] or no correlation [36] between hemoglobin and HbA_1c_. Recently, Sakamoto et al. demonstrated that elevated HbA_1c_ among anemic people with mean corpuscular volume (MCV) < 80 fL or MCV 80–90 fL, and decreased HbA_1c_ among anemic people with MCV > 90 fL, suggesting that different types of anemia might influence HbA_1c_ differently [32]. The inconsistent findings made it difficult to draw any conclusion regarding the influence of hemoglobin on HbA_1c_. In the present study, baseline hemoglobin was not associated with baseline HbA_1c_, but longitudinal change of hemoglobin had a positive correlation with HbA_1c_ longitudinal change after multivariable analysis. The temporal association between hemoglobin and HbA_1c_ over time might be more informative than that found in the cross-sectional studies. Hence, hemoglobin elevation with time might cause HbA_1c_ to increase concurrently. Previous study reported that genetic variation potentially affects HbA_1c_. Wheeler E. et al. [37] used genome-wide association meta-analyses in up to 159,940 individuals from 82 cohorts of European, African, East Asian, and South Asian ancestry. They identified 60 common genetic variants associated with HbA_1c_. Nineteen glycemic and twenty-two erythrocytic variants were associated with HbA1c at genome-wide significance. However, in our study, is it difficult to assess the influence of genetic variation on HbA1c, which represents a technical limitation. 

Baseline GOT was negatively associated with baseline HbA_1c_, but longitudinal change of GOT had no significant correlation with HbA_1c_ longitudinal change after multivariable analysis. Hence, longitudinal change of GOT may only have a minor role in HbA_1c_ longitudinal change (or no role at all). However, baseline GPT was positively associated with baseline HbA_1c_ and longitudinal change of GPT also had a positive correlation with HbA_1c_ longitudinal change in the multivariable analysis. Previous studies have demonstrated a relationship between elevated liver enzymes and incident type 2 diabetes [38,39]. In fact, such relationship was still obvious even when liver enzymes were within the normal range [40]. It suggested that liver enzymes were providing an indicator of liver fat content, which was linked to a constellation of metabolic disorders including hepatic insulin resistance, dyslipidemia, and obesity [15]. In the present study, we consistently found GPT change over time had a positive association with HbA_1c_ change over time. Hence, lowering GPT should be helpful in slowing HbA_1c_ longitudinal change.

Reduced albumin was reported to be associated with an unfavorable metabolic profile, characterized by increased adipose tissue inflammation, adiposity, and glucose and with an increased risk for type 2 diabetes mellitus [41]. In this study, we consistently found baseline albumin was negatively associated with baseline HbA_1c_ and longitudinal change of albumin also had a negative correlation with HbA_1c_ longitudinal change in the multivariable analysis. Hence, decreased albumin with time might result in HbA_1c_ increase concomitantly.

## 5. Study Limitations

There were several limitations to this study. First, our study population were selected from TWB and no medication data were available in this data bank. We excluded patients with diabetes mellitus s most of them received anti-diabetic mediation therapy, which certainly had an impact on the longitudinal change of HbA_1c_. Hence, our results could not apply to diabetic patients. Second, the information of anti-hypertensive medication and lipid lowering agents were similarly lacking. These medications undoubtedly influenced the values of blood pressures, resting heart rate, and lipid profile. Additionally, information about diet during the follow-up, anemia-related data including mean corpuscular volume, mean corpuscular hemoglobin, mean corpuscular hemoglobin concentration, serum iron, transferrin, and ferritin were lacking, which might influence the interpretation. Therefore, we could not exclude the impact of such medication on our present results. Third, according to the statistics of TWB, the proportion of participants coming back to track is about 50%. Therefore, selection bias may affect the interpretation of our results. Finally, we had no information concerning smoking status and exercise habit, so we could not analyze the impact of these parameters on the longitudinal change of HbA_1c_.

## 6. Conclusions

In this study including 23,326 subjects with non-diabetes, we found new-onset hypertension and longitudinal changes of resting heart rate, BMI, fasting glucose, total cholesterol, hemoglobin, and GPT were positively and longitudinal changes of serum creatinine and albumin were negatively correlated with HbA_1c_ longitudinal change in the multivariable analysis. Hence, strategies to decrease the development of new-onset hypertension, resting heart rate, BMI, fasting glucose, total cholesterol, and GPT and increase serum albumin level might be helpful in practice for slowing the longitudinal change of HbA_1c_ in non-diabetic population. In addition, increased hemoglobin and decreased serum creatinine over time also had an impact on the HbA_1c_ elevation over time in non-diabetic population.

## Figures and Tables

**Table 1 jpm-11-00648-t001:** Longitudinal changes of clinical and laboratory data during 3.8-year follow-up in 23,326 study subjects.

Parameters	Baseline	Follow-Up	*p* Value	Longitudinal Change
Age (year)	50.3 ± 10.4	54.2 ± 10.3	<0.001	3.8 ± 1.2
Hypertension (%)	10	14	<0.001	4
Systolic blood pressure (mmHg)	116 ± 17	123 ± 19	<0.001	7 ± 14
Diastolic blood pressure (mmHg)	72 ± 11	74 ± 11	<0.001	2 ± 9
Heart rate (beat/min)	69 ± 9	70 ± 9	<0.001	1.2 ± 9.0
Body mass index (kg/m^2^)	23.8 ± 3.4	24.1 ± 3.5	<0.001	0.30 ± 1.27
Fasting glucose (g/dL)	91.7 ± 7.3	92.3 ± 7.7	<0.001	0.60 ± 6.74
HbA_1c_ (%)	5.55 ± 0.33	5.67 ± 0.33	<0.001	0.10 ± 0.27
Creatinine (mg/dL)	0.71 ± 0.25	0.72 ± 0.31	<0.001	0.01 ± 1.16
Uric acid (mg/dL)	5.42 ± 1.41	5.37 ± 1.39	<0.001	−0.04 ± 0.92
Total cholesterol (mg/dL)	196 ± 35	198 ± 35	<0.001	2.2 ± 28.7
Triglyceride (mg/dL)	107 ± 73	114 ± 79	<0.001	6.5 ± 68.3
Red blood cell (*10^6^/μL)	4.72 ± 0.50	4.70 ± 0.51	<0.001	−0.01 ± 0.26
Hemoglobin (g/dL)	13.68 ± 1.55	13.67 ± 1.55	0.085	−0.01 ± 1.01
Hematocrit (%)	43.00 ± 4.46	40.96 ± 3.95	<0.001	−2.05 ± 3.27
GOT (μ/L)	24.1 ± 11.0	25.6 ± 11.3	<0.001	1.5 ± 12.8
GPT (μ/L)	22.4 ± 18.0	22.9 ± 19.5	0.002	0.45 ± 21.6
Albumin (g/dL)	4.55 ± 0.23	4.47 ± 0.22	<0.001	−0.08 ± 0.21

Abbreviations: HbA_1c_, Glycated Hemoglobin; GOT, glutamic-oxalocetic transaminase; GPT, glutamic-pyruvic transaminase.

**Table 2 jpm-11-00648-t002:** Univariable and multivariable correlates of baseline HbA_1c_.

Baseline Parameters	Baseline HbA_1c_			
	Univariable Analysis		Multivariable Analysis	
	β	*p*	β	*p*
Age (per 1 year)	0.322	<0.001	0.207	<0.001
Sex (male vs. female)	0.034	<0.001	−0.043	<0.001
Hypertension	0.109	<0.001	0.001	0.887
Systolic blood pressure (per 1 mmHg)	0.195	<0.001	−0.019	0.060
Diastolic blood pressure (per 1 mmHg)	0.143	<0.001	0.013	0.180
Heart rate (per 1 beat/min)	0.002	0.712	－	－
Body mass index (per 1 kg/m^2^)	0.194	<0.001	0.096	<0.001
Fasting glucose (per 1 g/dL)	0.378	<0.001	0.288	<0.001
Creatinine (per 1 mg/dL)	0.025	<0.001	−0.030	<0.001
Uric acid (per 1 mg/dL)	0.128	<0.001	0.027	<0.001
Total cholesterol (per 1 mg/dL)	0.225	<0.001	0.128	<0.001
Triglyceride (per 1 mg/dL)	0.153	<0.001	0.031	<0.001
Hemoglobin (per 1 g/dL)	0.092	<0.001	−0.007	0.366
GOT (per 1 U/L)	0.093	<0.001	−0.036	<0.001
GPT (per 1 U/L)	0.099	<0.001	0.068	<0.001
Albumin (per 1 g/dL)	−0.043	<0.001	−0.044	<0.001

Β, standardized coefficient; other abbreviations as in Table 1.

**Table 3 jpm-11-00648-t003:** Correlation between HbA_1c_ longitudinal change and longitudinal changes of other parameters in the univariable and multivariable analyses.

Longitudinal Changes of Parameters (△Parameters)	HbA_1c_ Longitudinal Change (△HbA_1c_)			
	Univariable Analysis		Multivariable Analysis	
	β	*p*	β	*p*
△Age (per 1 year)	0.022	0.001	−0.010	0.132
New-onset hypertension	0.013	0.048	0.014	0.028
△Systolic blood pressure (per 1 mmHg)	−0.009	0.166	－	－
△Diastolic blood pressure (per 1 mmHg)	0.046	<0.001	0.001	0.846
△Heart rate (per 1 beat/min)	0.037	<0.001	0.020	0.002
△Body mass index (per 1 kg/m^2^)	0.204	<0.001	0.171	<0.001
△Fasting glucose (per 1 g/dL)	0.135	<0.001	0.107	<0.001
△Creatinine (per 1 mg/dL)	−0.042	<0.001	−0.042	<0.001
△Uric acid (per 1 mg/dL)	0.041	<0.001	0.005	0.493
△Total cholesterol (per 1 mg/dL)	0.072	<0.001	0.040	<0.001
△Triglyceride (per 1 mg/dL)	0.058	<0.001	−0.002	0.846
△Hemoglobin (per 1 g/dL)	0.079	<0.001	0.062	<0.001
△GOT (per 1 U/L)	0.045	<0.001	−0.012	0.304
△GPT (per 1 U/L)	0.066	<0.001	0.041	0.001
△Albumin (per 1 g/dL)	−0.042	<0.001	−0.070	<0.001

Β, standardized coefficient; other abbreviations as in Table 1.

## Data Availability

The data underlying this study is from the Taiwan Biobank. Due to restrictions placed on the data by the Personal Information Protection Act of Taiwan, the minimal data set cannot be made publicly available. Data may be available upon request to interested researchers. Please send data requests to: Szu-Chia Chen, PhD, MD. Division of Nephrology, Department of Internal Medicine, Kaohsiung Medical University Hospital, Kaohsiung Medical University.

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
