# Peer review of "Determinants of Longitudinal Change of Glycated Hemoglobin in a Large Non-Diabetic Population"

_jpm, 2021, doi:10.3390/jpm11070648_

Round 1

Reviewer 1 Report

Well written abstract. Spacing issue noted on the method session. "Details regarding the TWB can be found from its official website (https://tai-wanview.twbiobank.org.tw/index)"

Author Response

Reviewer 1

Well written abstract. Spacing issue noted on the method session. "Details regarding the TWB can be found from its official website (https://tai-wanview.twbiobank.org.tw/index)"

Ans: Thank you for your suggestion. We have managed the problem.

Details regarding the laboratory data collection and questionnaire of TWB English version can be found from its official website (https://www.twbiobank.org.tw/new_web_en/). (Page 2, Line 42-44)

Reviewer 2 Report

The manuscript submitted by Ho-Ming Su et al. identifies the major determinants of longitudinal change of HbA1c in non-diabetic patients over a mean of 3.8 years of follow-up. This paper improves the understanding of the importance of HbA1c in non-diabetic patients. Although the data and the discussion are clearly presented, some questions following need to be addressed to strengthen their manuscript as follows:

1) Although authors excluded patients with diabetes at baseline or follow-up, the definition of people without diabetes are fasting glucose <126 mg/dl and HbA1c <6.5%. In this case, the object of the study contained both normal and pre-diabetes (impaired fasting glucose: IFG). If the group is divided into 2 groups (group of FPG<100 and group of 100<FPG<126), are there any differences observed? Does that affect the conclusion?

2) As named Glycated hemoglobin (HbA1c), a form of hemoglobin, it was impacted by red blood cell lifespan and the enzymes regulate glycemic pathways. For instance, HbA1c tends to falsely lower in the patient with hemolytic anemia due to reduced erythrocyte lifespan whereas HbA1c is known to raise in the case of iron deficiency. Recently it was reported that genetic variation potentially affects HbA1c (see the References). They identified 60 genetic variants among European, African, East Asian, and South Asian ancestry. Among them, 19 were classified as influencing HbA1c through glycemic pathways, 22 through erythrocytic pathways. 

It is difficult to examine the whole sequence of 27,209 participants. However, it is better to discuss this issue (in the case authors cannot examine whole, discuss as a technical limitation) in the discussion part. In addition, It is better to add the information of anemia-related data including Hb, Ht, RBC, MCV, MCH, MCHC, Serum iron, transferrin ferritin etc as much as possible even in the supplemental figure/table.

References

- Impact of common genetic determinants of hemoglobin A1C on type 2 diabetes risk and diagnosis in ancestrally diverse populations: a transethnic genome-wide meta-analysis. PLoS Med, 14 (2017), p. e1002383

- Genetics of HbA1c: a case study in clinical translation. Current Opinion in Genetics & Development 50, (2018), 79-85 

Author Response

Reviewer 2

The manuscript submitted by Ho-Ming Su et al. identifies the major determinants of longitudinal change of HbA1c in non-diabetic patients over a mean of 3.8 years of follow-up. This paper improves the understanding of the importance of HbA1c in non-diabetic patients. Although the data and the discussion are clearly presented, some questions following need to be addressed to strengthen their manuscript as follows:

  1. Although authors excluded patients with diabetes at baseline or follow-up, the definition of people without diabetes are fasting glucose <126 mg/dl and HbA1c <6.5%. In this case, the object of the study contained both normal and pre-diabetes (impaired fasting glucose: IFG). If the group is divided into 2 groups (group of FPG<100 and group of 100<FPG<126), are there any differences observed? Does that affect the conclusion?

Ans: Thank you for your comments. We have further divided the study participants into 2 groups (group of FPG <100 mg/dL and group of 100 ≤ FPG < 126 mg/dL), and found the similar results. We have added the data in the results.

  • We have further performed two subgroup analyses (group of fasting glucose <100 mg/dL and group of 100 ≤ fasting glucose < 126 mg/dL) to survey the correlation between HbA1c longitudinal change and longitudinal changes of other parameters. In participants with fasting glucose <100 mg/dL (n = 20,248), new-onset hypertension (coefficient β: 0.018, p = 0.039), low △systolic blood pressure (coefficient β: -0.001, p < 0.001), high △diastolic blood pressure (coefficient β: 0.001, p < 0.001), high △BMI (coefficient β: 0.034, p < 0.001), high △fasting glucose (coefficient β: 0.005, p < 0.001), low △creatinine (coefficient β: -0.077, p < 0.001), high △total cholesterol (coefficient β: 0, p < 0.001), high △hemoglobin (coefficient β: 0.014, p < 0.001), high △GPT (coefficient β: 0, p = 0.012), and low △albumin (coefficient β: -0.082, p < 0.001) were significantly associated with high △HbA1c.
  • In participants with 100 ≤ fasting glucose < 126 mg/dL (n = 3,078), high △heart rate (coefficient β: 0.001, p = 0.009), high △BMI (coefficient β: 0.053, p < 0.001), high △fasting glucose (coefficient β: 0.004, p < 0.001), low △creatinine (coefficient β: -0.049, p = 0.036), low △uric acid (coefficient β: -0.009, p = 0.040), high △total cholesterol (coefficient β: 0, p = 0.015), high △hemoglobin (coefficient β: 0.029, p < 0.001), low △GOT (coefficient β: -0.001, p = 0.045), high △GPT (coefficient β: 0.001, p = 0.049), and low △albumin (coefficient β: -0.124, p < 0.001) were significantly associated with high △HbA1c. (Page 3, under Table 3)
  1. As named Glycated hemoglobin (HbA1c), a form of hemoglobin, it was impacted by red blood cell lifespan and the enzymes regulate glycemic pathways. For instance, HbA1c tends to falsely lower in the patient with hemolytic anemia due to reduced erythrocyte lifespan whereas HbA1c is known to raise in the case of iron deficiency. Recently it was reported that genetic variation potentially affects HbA1c (see the References). They identified 60 genetic variants among European, African, East Asian, and South Asian ancestry. Among them, 19 were classified as influencing HbA1c through glycemic pathways, 22 through erythrocytic pathways. It is difficult to examine the whole sequence of 27,209 participants. However, it is better to discuss this issue (in the case authors cannot examine whole, discuss as a technical limitation) in the discussion part. In addition, It is better to add the information of anemia-related data including Hb, Ht, RBC, MCV, MCH, MCHC, Serum iron, transferrin ferritin etc as much as possible even in the supplemental figure/table.

References

  1. Impact of common genetic determinants of hemoglobin A1C on type 2 diabetes risk and diagnosis in ancestrally diverse populations: a transethnic genome-wide meta-analysis. PLoS Med, 14 (2017), p. e1002383
  2. Genetics of HbA1c: a case study in clinical translation. Current Opinion in Genetics & Development 50, (2018), 79-85

Ans: Thank you for your comments. We have added this issue in the Discussion. Besides, we have added the data of RBC, Hgb and Hct in Table 1 (we do not have the data of MCV, MCH, MCHC, Serum iron, transferrin and ferritin). We had added Hgb in multivariable analysis in Table 2 and 3 at last version. We have put the issue about lacking anemia-related data in the Limitation.

  • Previous study reported that genetic variation potentially affects HbA1c. Wheeler E et al.[37] used genome-wide association meta-analyses in up to 159,940 individuals from 82 cohorts of European, African, East Asian, and South Asian ancestry, they identified 60 common genetic variants associated with HbA1c. Nineteen glycemic and 22 erythrocytic variants were associated with HbA1c at genome-wide significance. However, in our study, is it difficult to assess the influence of genetic variation on HbA1c, which is a technical limitation. (Page 7, Line 9-16)
  • Besides, information about diet during the follow-up, anemia-related data including mean corpuscular volume, mean corpuscular hemoglobin, mean corpuscular hemoglobin concentration, serum iron, transferrin, and ferritin were lacking, which might influence the interpretation. (Page 7, Line 44-48)

Reviewer 3 Report

  • In "Study limitations", add the lack of information about diet during the follow-up.
  • Consider that the changes, although significant, are very small (for instance 1 beat/min for heart rate). This must be commented.
  • I wonder if decrease resting heart rate, GPT and increase serum albumine may be helpful in practice for showing the longitudinal change of HbA1C.
  • Table 1: Correct in the "Baseline" column / line "Body mass index", delete "N".

Author Response

Reviewer 3

  1. In "Study limitations", add the lack of information about diet during the follow-up.

Ans: Thank you for your comments. We have added this issue in the Limitation.

  • Besides, information about diet during the follow-up, anemia-related data including mean corpuscular volume, mean corpuscular hemoglobin, mean corpuscular hemoglobin concentration, serum iron, transferrin, and ferritin were lacking, which might influence the interpretation. (Page 7, Line 44-48)
  1. Consider that the changes, although significant, are very small (for instance 1 beat/min for heart rate). This must be commented.

Ans: Thank you for your comments. We totally agreed your point. We have added the sentences in the Discussion.

  • However, although significant, the HbA1c longitudinal change was small (0.020%) when heart rate increased by 1 beat/min. (Page 6, Line 8-9)
  1. I wonder if decrease resting heart rate, GPT and increase serum albumin may be helpful in practice for showing the longitudinal change of HbA1C.

Ans: Thank you for your comments. We have revised our sentence.

  • Hence, strategies to decrease the development of new-onset hypertension, resting heart rate, BMI, fasting glucose, total cholesterol, and GPT and increase serum albumin level might be helpful in practice for slowing the longitudinal change of HbA1c in non-diabetic population. (Page 8, Line 6-9)
  1. Table 1: Correct in the "Baseline" column / line "Body mass index", delete "N".

Ans: Thank you for your correction. We have deleted “N”.